# Impact of Cryogenic Treatment Process on the Performance of 51CrV4 Steel

**DOI:** 10.3390/ma16124399

**Published:** 2023-06-15

**Authors:** Zhi Chen, Linwang Jing, Yuan Gao, Yao Huang, Jia Guo, Xianguo Yan

**Affiliations:** 1School of Mechanical Engineering, Taiyuan Science and Technology University, Taiyuan 030024, China; mechenzhi@tyust.edu.cn (Z.C.); hy343999539@163.com (Y.H.); 1983030@tyust.edu.cn (X.Y.); 2School of Mechanical Engineering and Automation, Shanghai University, Shanghai 200444, China; sdgaoyuan2020@163.com; 3Technical Insitute of Physics and Chenmistry, Beijing 100190, China; guojia@mail.ipc.ac.cn

**Keywords:** gray relational analysis, cryogenic treatment, impact toughness, wear resistance

## Abstract

The working load on automotive components is continuously rising, and the mechanical performance requirements for component materials are rising along with the growth trend toward light weight and high dependability in automobiles. In this study, the response characteristics of 51CrV4 spring steel were taken to be its hardness, wear resistance, tensile strength, and impact toughness. Prior to tempering, cryogenic treatment was introduced. Through the Taguchi method and gray relational analysis, the ideal process parameters were discovered. The ideal process variables were the following: a cooling rate of 1 °C/min, a cryogenic temperature of −196 °C, a holding time of 24 h, and a cycle number of three. An analysis of variance revealed that the holding time had the greatest effect on the material properties, with an effect of 49.01%. The yield limit of 51CrV4 was increased by 14.95% and the tensile strength was increased by 15.39% with this group of processes, and the wear mass loss was reduced by 43.32%. The mechanical qualities had a thorough upgrade. Microscopic analysis revealed that cryogenic treatment resulted in refinement of the martensite structure and significant differences in orientation. Additionally, bainite precipitation occurred, exhibiting a fine needle-like distribution, which positively influenced impact toughness. Analysis of the impact fracture surface showed that cryogenic treatment led to an increase in dimple diameter and depth. Further analysis of the elements revealed that calcium (Ca) weakened the negative effect of sulfur (S) on 51CrV4 spring steel. The overall improvement in material properties provides guidance for practical production applications.

## 1. Introduction

Widely used in the automotive industry, 51CrV4 steel is a medium carbon Cr-Mn-V alloy spring steel with applications such as suspension coil springs, torsion springs, and variable cross-section leaf springs [1,2,3]. To meet the increasing demands for higher mechanical performance in the automotive industry, which emphasizes lightweight design and enhanced reliability, it is crucial to address the limitations of conventional heat treatment methods for steel. These methods often result in non-uniform or insufficient heating, leading to the presence of significant retained austenite. The instability of retained austenite can result in dimensional inaccuracies in workpieces and negatively impact fatigue strength and other material properties [4]. Recent studies have shown that through incorporating cryogenic treatment before tempering, significant improvements can be achieved. Cryogenic treatment has been found to reduce the residual austenite content after heat treatment [5,6]. It also helps regulate residual stresses [7], refine the microstructure of the material, improve its properties, and extend the service life of the workpiece [8]. By enhancing existing processes and adopting cryogenic treatment, it is possible to overcome the limitations of conventional heat treatment methods and optimize the performance of 51CrV4 steel.

Shallow and deep cryogenic treatments utilize liquid nitrogen as a cooling medium to cool the material to a specific subzero temperature before tempering [9]. These treatments have been shown to enhance various material properties, including increased hardness and wear resistance, extended service life, reduced residual stress, and improved electrical conductivity [10,11,12,13,14,15]. In a study conducted by R.C. Özden et al. [16], different process treatments for EN52CrMoV4 spring steel were investigated. The results indicate that direct deep cryogenic treatment after quenching, followed by tempering, resulted in smaller grain size and the precipitation of ultrafine carbides (M2C). As a result, the material exhibited simultaneous improvements in fatigue life, ductility, and strength. Similarly, Vinícius Richieri Manso Gonçalves et al. [2] conducted multiple experiments and found that deep cryogenic treatment had minimal influence on the hardness of spring steel. Furthermore, the fracture toughness of various spring steels was compared after deep cryogenic treatment, with 51CrV4-ACF steel showing the greatest improvement in performance. In a study by Fuat Kara et al. [17], deep cryogenic treatment was applied to Sleipner cold work tool steel at different temperatures. It was observed that the treatment resulted in finer and more uniformly distributed carbide particles. Friction coefficient measurements revealed that low-temperature treatment reduced the friction coefficient and improved the wear resistance of the material. M. Arun et al. [18] conducted experiments on tungsten carbide drill bits at different deep cryogenic temperatures and found that deep cryogenic treatment (−196 °C) was necessary for tooling during the manufacturing of long series of holes. This treatment increased the hardness of the tool, resulting in lower thrust and reducing the roughness and roundness errors of the workpiece. Haidong Zhang et al. [19] designed an orthogonal experiment to investigate deep cryogenic treatment combined with low-temperature tempering on 42CrMo steel. They observed a significant improvement in wear resistance after deep cryogenic treatment and found that the deep cryogenic treatment had the greatest impact, contributing 47.35% to the overall improvement. Apart from the temperature of the cold treatment, other parameters during the cold treatment stage also have significant effects on material performance. Haidong Zhang et al. [20] utilized an orthogonal design approach to investigate the deep cryogenic cycling treatment of 42CrMo steel. They found that for optimizing wear resistance, a cycling frequency of two cycles was the most effective, while for optimizing impact toughness, a cycling frequency of three cycles was optimal. Varying the number of deep cryogenic cycles resulted in different material properties. Guili Xu et al. [21] observed the microstructural changes in AISI M35 high-speed steel treated at −196 °C for different insulation times. They determined that the optimal deep cryogenic treatment time was 5 h, as increasing the time had minimal influence on material performance. Lalithkumar Bhaskar et al. [22] presented different findings regarding deep cryogenic temperature. Their study indicates that AISI316-SS drills exhibited the best performance when treated for 24 h. G. Mazor et al. [23] investigated the impact of two cooling methods on the hardness and wear resistance of various tool steels, concluding that rapid cooling resulted in improved material performance. Honglin Zhang et al. [24] conducted cold treatments on M54 secondary hardening steel using different cooling rates and compared the final performances. They found that a cooling rate of 3 °C/min achieved the best performance for the material.

Cryogenic treatment induces lattice contraction, leading to an increase in dislocation density and the precipitation of carbide particles. These alterations modify the material’s microstructure and have a significant impact on its macroscopic performance. P.H.S. Cardoso et al. [25] conducted deep cryogenic treatment experiments on AISI D6 tool steel and found that with the extension of holding time, the hardness of the material increased. After deep cryogenic treatment and tempering, the secondary carbide particles increased, leading to higher hardness and impact toughness of the material without sacrificing its wear resistance. N.B. Dhokey et al. [26] conducted experiments on M35 tool steel at −185 °C with different holding times and observed an increasing trend of carbide particle coarsening with increasing holding time. The shape of the carbides changed from spherical to near-angular, which caused the strain in the matrix. Deep cryogenic treatment stores energy in the matrix in the form of residual stress. The excessive accumulation of residual stress resulted in a shear fracture. The effects of cryogenic treatment on the microstructure of materials are reflected in the reduction in grain size at low temperatures, the promotion of the formation and uniform distribution of small secondary carbides, the increase in the number of carbides and the volume fraction of martensite in the matrix, and the decrease in their average particle spacing [27,28]. Cryogenic treatment accelerates the production of a large number of small second-phase precipitates at low temperatures [1].

When considering multiple factors and levels for a single response characteristic, the Taguchi method is commonly used to efficiently obtain experimental data and ensure data reliability. However, when optimizing multiple response characteristics, the Taguchi method may struggle to provide the most objective results. Gray relational analysis effectively correlates complex relationships among multiple performance characteristics [29]. Derzija Begic-Hajdarevic et al. [30] employed a Taguchi-based gray relational analysis approach to comprehensively consider parameters such as feed rate, cutting speed, drill type, and drill angle and optimize multiple performance indicators such as surface roughness and burr height, thus determining the optimal machining conditions. Jayakrishnan Unnikrishna Pillai et al. [31] applied Taguchi gray relational analysis to optimize process parameters for a six-axis robotic machining center, specifically a vertical milling machine. They obtained the optimal process combination and determined, through variance analysis, that the tool path strategy had the greatest impact on performance characteristics such as machining time and surface roughness. J.D. Darwin et al. [32] utilized the Taguchi method to optimize the processing parameters of a low-temperature treatment, aiming to enhance the wear resistance of 18% Cr martensitic stainless steel. They determined the most significant parameter affecting wear resistance to be the immersion temperature, which was found to be −184 °C. The immersion temperature was found to contribute approximately 72% to the overall improvement in wear performance. The combination of the Taguchi method and gray relational analysis allows for the determination of optimal processes and the optimization of multiple performance indicators. This combined approach is particularly useful in multi-objective optimization.

Currently, there is limited research on the cryogenic treatment of 51CrV4 spring steel, and studies on other spring steels have only focused on a single low-temperature treatment cycle. This study employed a combination of gray Taguchi experiments to optimize the cryogenic treatment parameters of 51CrV4 spring steel with the aim of enhancing its hardness, wear resistance, and impact toughness. The optimized cryogenic treatment process was designed to ensure that the material’s strength remained uncompromised. Additionally, a detailed analysis was conducted to investigate the influence of cryogenic treatment on the microstructure of the material. Importantly, the study maximized the material’s performance without altering its elemental composition, providing valuable guidance for practical production applications.

## 2. Experimental Procedure

### 2.1. Material Selection and Temperature Settings

The material selected in this experiment was 51CrV4 spring steel, and the chemical composition analyzed via the spectrometer (QSN 750-II, OBLF, Witten, Germany) is shown in Table 1.

Taking into account the influence of four factors, including cryogenic temperature, holding time, cooling rate, and cryogenic cycle, on the wear resistance, hardness, and impact toughness of the material, an orthogonal experimental design table for cryogenic treatment (shown in Table 2) and a heat treatment program (shown in Figure 1) were designed. All the specimens in this experiment were held at 900 °C for 0.5 h, then held in a fast bright quenching oil (E-CH01, Yingji, China) at 70 °C for 0.5 h, and then removed and cooled to room temperature in air. Within 24 h, the materials prepared were subjected to cryogenic treatment as shown in Table 2. After the cryogenic treatment, the specimens were tempered within 24 h at a temperature of 480 °C for 1 h. After tempering, the specimens were cooled automatically to room temperature by closing the furnace door in the furnace.

### 2.2. Testing of Hardness and Wear Volume

The specimens were prepared as cylindrical discs with a diameter of Φ25 mm and a thickness of 10 mm for testing hardness and wear resistance. The Rockwell hardness of the 51CrV4 spring steel was measured using a Rockwell hardness tester (HR-150A, Huayin, China). In the study, all specimens were measured under the same conditions. Each experimental group consisted of three specimens, and five points were measured on each specimen. After excluding the maximum and minimum values, the average value was taken as the hardness of the specimens.

During the investigation of the impact of cryogenic treatment on the wear resistance of 51CrV4 steel, the design of the rotating friction parameters was carried out according to Table 3 using a friction tester (CTF-I, Kaihua, China). The transient coefficient of friction during the wear test was automatically measured and recorded by the device. The wear volume of the specimens was measured using an analytical balance with a range of 10 mg to 200 g and a readability of 0.1 mg. The measurement of specimen mass was the average value obtained from multiple measurements.

### 2.3. Testing of Impact Toughness of 51CrV4 Spring Steel

The impact toughness tests were conducted according to the ASTM E23-12C standard, using Charpy V-notch specimens with dimensions as shown in Figure 2. The Charpy V-notch specimens were selected for impact toughness testing, and the specimen dimensions are shown in Figure 2. During specimen preparation, the ports were tightly wrapped with tin foil and tape to prevent cooling fluid from splashing onto them. The impact energy of all specimens was tested at room temperature using an impact testing machine (JB-300B, Tianchen, Tianjin, China), and the impact energy was converted into impact toughness values. After the samples were cut, their surfaces were roughly polished and then polished again using a polishing machine. They were subsequently subjected to corrosion using a HNO3:C2H5OH = 1:24 (vol.) solution, and their microstructure was observed using a scanning electron microscope (JSM-6510, JEOL, Tokyo, Japan).

### 2.4. Tensile Testing of 51CrV4 Spring Steel

The tensile specimens were designed in accordance with the “dog bone” shape specified by GB/T228.1-2021 standard, as shown in Figure 3. They were tested on a microcomputer-controlled electronic universal testing machine (WDW-10D Fengzhi, China) at a pulling speed of 0.0010 mm/min. There were a total of 6 tensile specimens, with 3 specimens only undergoing quenching and tempering processes and the other 3 specimens undergoing the optimal treatment process to test the effect of optimal cryogenic treatment on the yield strength and tensile strength of 51CrV4 spring steel. The tensile tests were conducted in the same environment and within the same time frame to ensure the reliability of the test results.

## 3. Results and Discussion

### 3.1. Experimental Data

#### 3.1.1. Experimental Data Preprocessing

In order to assess the impact of various cryogenic parameters on the performance of 51CrV4 spring steel, a series of experiments were conducted, focusing on wear resistance, hardness, and impact toughness as the response characteristics. The obtained results are presented in Table 4. Subsequently, range analysis was performed on the experimental data, and the outcomes of the range analysis are depicted in Table 5. Based on the range analysis, it was observed that factors A (cooling rate), B (cryogenic temperature), C (holding time), and D (number of cycles) exhibited diverse effects on each performance index. The order of importance for the different cryogenic factors on wear resistance was determined to be A > C > D > B. Regarding hardness, the order of importance was C > A > D > B, while for impact toughness, the order of importance was D > C > A > B.

Based on the range analysis, it was evident that the factors exerting the most significant influence on wear resistance, hardness, and impact toughness were not consistent. To address this issue, gray correlation analysis was employed to optimize the multi-objective data. The measured values of wear, hardness, and impact energy were obtained in different ranges and units, which posed challenges for data analysis. To enhance data comparability, a preprocessing step was performed. The objective of enhancing the performance of 51CrV4 steel encompassed reducing wear, improving hardness, and increasing impact toughness. Therefore, the experimental data were normalized using the minimum wear, maximum hardness, and maximum impact energy as reference points, and the normalized data are presented in Table 6.

When maximizing the desired target output is preferred, the “the bigger, the better” criterion is used, and the normalization formula is as follows:(1)$$r_{\mathrm{ij}}=\frac{x_{\mathrm{ij}}-\underset{i}{\min x_{\mathrm{ij}}}}{\underset{i}{\max x_{\mathrm{ij}}}-\underset{i}{\min x_{\mathrm{ij}}}}$$

When minimizing the desired target output is preferred, the “the smaller, the better” criterion is used, and its normalization formula is shown below:(2)$$r_{\mathrm{ij}}=\frac{\underset{i}{\max x_{\mathrm{ij}}}-x_{\mathrm{ij}}}{\underset{i}{\max x_{\mathrm{ij}}}-\underset{i}{\min x_{\mathrm{ij}}}}$$

The experimental data were normalized using the principles of “the bigger, the better” and “the smaller, the better”. The larger the normalized value of each group of data, the closer the experimental result was to the desired outcome. A normalized value of 1 represented the best experimental result [29,30,31,32,33].

#### 3.1.2. Gray Correlation Analysis

The gray correlation coefficient, also known as the correlation degree, was used to evaluate the correlation between each parameter and the output target. A higher gray correlation coefficient indicates a stronger correlation with the output target. The formula for calculating the gray correlation coefficient is as follows:(3)$$\xi_{\mathrm{ij}}=\frac{\underset{i}{\min}\underset{j}{\min\Delta_{\mathrm{ij}}}+\gamma \underset{i}{\max}\underset{j}{\max\Delta_{\mathrm{ij}}}}{\Delta_{\mathrm{ij}}+\gamma \underset{i}{\max}\underset{j}{\max\Delta_{\mathrm{ij}}}}$$

In Formula (3), $\Delta_{\mathrm{ij}}$ was the mass loss function,$\;\Delta_{\mathrm{ij}}=\left|\bar{r_{\mathrm{ij}}}-r_{\mathrm{ij}}\right|$, $r_{ij}$ was the normalized target value of the i-th evaluation index data, and γ was the resolution coefficient, γ =0.5.

The gray correlation coefficients for different indicators, calculated using Equation (1), are presented in Table 7. Among the orthogonal experimental groups, group 1 exhibited the highest gray correlation coefficient for wear amount, group 7 showed the highest gray correlation coefficient for hardness, and group 6 displayed the highest gray correlation coefficient for impact resistance. These parameter combinations demonstrate the strongest correlation among the indicators. In evaluating the overall comprehensive performance, the overall gray correlation degree was considered. The magnitude of the overall gray correlation degree reflected the influence of corresponding factors on the material’s comprehensive performance. A larger overall gray correlation degree indicated a greater impact of the cryogenic process combination on the overall performance. The formula for calculating the overall gray correlation degree is as follows:(4)$$\beta_{i}=\frac{1}{n}\sum_{j=1}^{n}\xi_{\mathrm{ij}}$$

Based on the rankings provided in Table 7, it was determined that group 3 achieved the highest overall gray correlation degree of 0.72476, making it the optimal combination. The corresponding process parameters for this experiment were a cooling rate of 1 °C/min, a cryogenic temperature of −196 °C, a holding time of 24 h, and three cycles. This particular process configuration exhibited the best overall performance among the 10 conducted experiments. To visualize the changes in the overall gray correlation degree, a graph depicting the overall gray correlation degree was plotted and is presented in Figure 4.

According to the principle of orthogonal design, the optimal cryogenic treatment process could have been outside the 10 groups of experiments. Therefore, to obtain the optimal cryogenic treatment process, further analysis of the gray correlation degree of each group of experiments obtained from Table 7 was necessary. The average gray correlation degree could significantly demonstrate the correlation between each factor and the target. The larger the average gray correlation degree of the factor, the greater its influence on the target level. To further analyze the impact of the cryogenic process on the comprehensive performance of 51CrV4 spring steel, the average gray correlation degree of each factor level was calculated based on the overall gray correlation degree of the 10 experiments. The calculation results are shown in Table 8. The average gray correlation response of factors A (cooling rate), B (cryogenic temperature), C (holding time), and D (number of cycles) at three levels is shown in Figure 5.

Based on Table 8 and Figure 5, it was found that the average gray correlation degree was highest for the first level of cooling rate (1 °C/min), the third level of cryogenic temperature (−196 °C), the third level of holding time (24 h), and the second level of cryogenic cycles (2 times). Therefore, the optimal cryogenic treatment process combination was A1B3C3D2, which means the cooling rate was 1 °C/min, the cryogenic temperature was −196 °C, the holding time was 24 h, and the cryogenic cycles were performed twice.

#### 3.1.3. Analysis of Variance

Variance analysis was conducted on the three factors other than cryogenic temperature, as the previous analysis showed that cryogenic temperature had the least impact on the overall performance of 51CrV4 spring steel. The contribution of each factor to the experimental results was determined. The total sum of squares and total degrees of freedom are shown below:(5)$$SS_{j}=r\sum_{i=1}^{n}{\left(x_{i}-\bar{x}\right)}^{2}=\;\sum_{i=1}^{m}{x_{i}}^{2}-\frac{T^{2}}{n}$$
(6)$$f_{T}=n-1$$

The formulas for the deviation sum of squares and degrees of freedom for each column are as follows:(7)$$SS_{j}=r\sum_{i=1}^{m}{\left(t_{i}-\bar{x}\right)}^{2}=\;\frac{1}{r}\sum_{i=1}^{m}{T_{i}}^{2}-\frac{T^{2}}{n}(j=1,2,\ldots ,k)$$
(8)$$f_{j}=m-1$$

If factor A was arranged in the jth column (j = 1, 2, …, k) of the orthogonal table, then we could obtain $\mathrm{SSA}\;={SS}_{j}$, at this time the r = n/m. The formula for the error sum of squares and degrees of freedom is:(9)$$SSE=SST-\sum_{j=1}^{k}{SS}_{j}$$
(10)$$f_{e}=f_{T}-\sum_{j=1}^{k}f_{j}$$

The calculated results are presented in Table 9, revealing that the holding time made the highest contribution at 49.01%, followed by the cooling rate at 27.74%, while the number of cycles had the smallest contribution at 18.85%. The variance analysis results demonstrate that the significance ranking of the factors influencing the performance of 51CrV4 steel is as follows: holding time > cooling rate > number of cycles. This ranking aligns with the findings obtained from the gray correlation analysis.

### 3.2. Testing and Analysis of Wear Resistance

Based on the results obtained from the aforementioned gray correlation analysis, the process group with the best performance was identified as group 3, characterized by a cooling rate of 1 °C/min, a cryogenic temperature of −196 °C, a holding time of 24 h, and three cycles. The friction system utilized in the experiment is depicted in Figure 6. In terms of theoretical wear resistance testing, a smaller mass loss indicates superior wear resistance of the specimen. For the samples in this particular group, the mass loss of wear measured 6.866 mg, which corresponds to an approximate 43.32% reduction compared to the control group. The Rockwell hardness of the samples exhibited a slight improvement, maintaining a level comparable to that of the control group. Furthermore, the impact toughness of the samples increased by 4.01% in comparison to the control group. The friction and wear test also allowed for the direct determination of the average friction coefficient of the ceramic ball under the test conditions employed. In this study, we focused on comparing and analyzing the average friction coefficients between the samples in the optimized process group and the control group. The instantaneous friction coefficients of the samples from both process groups are illustrated in Figure 7.

Figure 7 illustrates the temporal variation in the friction coefficient during the experimental process. In the initial 7 min, the friction coefficient exhibited a gradual increase. This can be attributed to the uneven surface of the specimen during the initial stages of wear, resulting in a smaller contact radius in the friction area and a relatively limited wear area. Consequently, the friction coefficient increased rapidly. As time progressed, the surface of the specimen underwent further wear, leading to an enlargement of the contact area in the friction region. This, in turn, resulted in a decrease in the wear rate, and the friction coefficient began to stabilize. Eventually, the experiment reached a steady state, and the friction coefficient remained relatively constant. With the increase in wear time, the contact radius between the ceramic ball and the surface expanded, leading to an elevation in the intermediate contact pressure. Consequently, the smooth surface that had formed began to suffer damage. Concurrently, the surface roughness at the edge was relatively high, causing an increase in the instantaneous friction coefficient. Once the wear band no longer experienced destruction, the friction coefficient tended to stabilize. The cryogenic treatment group exhibited a relatively slow increase in the instantaneous friction coefficient due to the enhanced wear resistance resulting from increased material hardness [34]. The prolonged time for the destruction of the smooth band contributed to a slower overall trend change in the cryogenic treatment group compared to the control group. Figure 7 demonstrates that the instantaneous friction coefficients measured from specimens in both test groups exhibited fluctuation around a certain average value while maintaining an overall linear trend. Various factors influence the coefficient of friction during the wear process. As the wear time lengthens, the surface temperature of the material increases, which significantly impacts the coefficient of friction. With rising temperature, an oxide film forms on the specimen’s surface, acting as a solid lubricant and stabilizing the instantaneous friction coefficient within a certain range, thereby maintaining a relatively low value. However, prolonged friction disrupts the formation of the oxide film, causing the friction coefficient to gradually increase [35]. Once the formation rate of the oxide film and the disruption rate reach a relative stability, the friction coefficient tends to stabilize. Nonetheless, the overall trend of the friction coefficient fluctuation remains around its average value. Based on Figure 7, it can be inferred that the average coefficient of friction for group 3 was approximately 0.45, while the average coefficient of friction for the control group was about 0.58. Deep cryogenic treatment reduced the instantaneous friction coefficient of the material and improved its wear resistance.

### 3.3. Analysis of Tensile Test Results

The tensile mechanical properties of 51CrV4 spring steel after the optimal cryogenic treatment and conventional heat treatment are shown in Figure 8. The red line in Figure 8 represents an extension of the initial linear portion of the curve. By shifting this line 0.2% to the right, we obtain the blue line shown in Figure 8. At the point where the blue line intersects the stress–strain curve, we can determine the yield strength of the material. The yield strength of the conventionally heat-treated sample was about 1068.4 MPa, while the yield strength of the sample treated with the optimal cryogenic process was about 1233.5 MPa, an increase of 14.95%. The elongation at fracture was also improved, with the elongation at fracture of the conventionally heat-treated sample being 11.61% and the elongation at fracture increasing to 14.28% after cryogenic treatment. At the same time, the tensile strength was also increased by 15.39%, as shown in Figure 9. It could be concluded that deep cryogenic treatment effectively improved the tensile strength and yield strength of 51CrV4 spring steel.

Figure 10 depicts the tensile fracture surface of the samples from the control group. In Figure 10a, it is evident that the fracture surface of the control group sample exhibited small cleavage facets and fine dimples, displaying quasi-cleavage fracture characteristics. However, the fracture surface of the cryogenically treated sample exhibited fine dimples and tearing edges with significant deformation. Some of the dimples contained inclusions, and the size of these inclusions was relatively large, indicating a ductile fracture, as shown in Figure 10b. The specimens from the control group underwent only quenching and tempering processes. Traditional heat treatment resulted in the aggregation of carbides at grain boundaries, thereby weakening them. Consequently, cracks tended to propagate along the more brittle directions of the material, displaying intergranular fracture characteristics. In contrast, the deep cryogenic treatment process reduced the grain size and led to finer and more uniformly distributed carbide particles within the grains, thus mitigating their detrimental effect on the grain boundaries [21,25,26,27,28,36]. During tensile testing, carbide particles became the starting source of toughness dimples, and the uniform distribution of the carbides made the distance between them closer, making it easier for toughness dimples to coalesce. Consequently, 51CrV4 spring steel exhibited better plasticity.

### 3.4. Analysis of Microstructure

In order to investigate the microstructural changes in 51CrV4 spring steel after different cryogenic treatments in comparison to the control group specimens, scanning electron microscopy (SEM) was employed to observe the specimens prepared for each experimental group. Based on the impact of holding time, the discussions were categorized accordingly, as the holding time was identified as the main factor influencing the comprehensive performance according to the gray correlation analysis. Microstructure images of specimens held for 2 h are presented in (a), (f), and (h) in Figure 11, while specimens held for 12 h are displayed in (b), (d), and (i). Additionally, specimens held for 24 h are shown in (c), (e), and (g).

Figure 11a reveals the presence of numerous short and coarse martensite structures. It should be noted that the test temperature in the first cryogenic treatment group was −120 °C, which was not sufficiently low, and the holding time was only 2 h, leading to an incomplete transformation of retained austenite. As a result, some residual austenite precipitated as the final structure, which exhibited lower hardness [37]. While the transformation of martensite contributes to improvement in material hardness, the presence of short and coarse martensite structures, along with the precipitation of residual austenite, contributed to poor hardness performance. This is an important factor contributing to the observed lower impact toughness and Rockwell hardness in this particular group of specimens [38]. In the microstructure image of group 6, as depicted in Figure 11f, the prominent characteristic is the presence of fine needle-like bainite, which significantly enhances the material’s impact toughness. However, the transformation of martensite in this group is limited, resulting in a reduction in hardness [39]. Furthermore, the absence of noticeable carbide particles in this particular group is a critical factor contributing to its inferior wear resistance. The microstructure image of group 8, as shown in Figure 10h, exhibited similarities to that of group 6. It revealed the presence of regionally distributed needle-like bainite, characterized by a dense and uniform structure. Conversely, the quantity of short rod-like martensite was limited and coarse, interspersed within the bainite matrix. Notably, this microstructure configuration demonstrated excellent impact toughness while exhibiting relatively low hardness. Consequently, it can be concluded that increasing the number of cryogenic cycles while maintaining the same 2-h holding time facilitated the formation of dense needle-like bainite structures and resulted in improved impact toughness.

The microstructure of group 2 is illustrated in Figure 11b. This group of specimens displayed a distinct microstructural feature characterized by the presence of numerous fine needle-like bainite structures, with particle-like carbides observed on the surface. Notably, these specimens exhibited excellent wear resistance and impact toughness. Furthermore, the microstructure of group 4, as depicted in Figure 11d, revealed an even greater prevalence of finely-needled bainite precipitation in comparison to group 2. The microstructure of group 4 exhibited sporadic distribution of short, thick rod-shaped bainite within the dense needle-shaped bainite, which significantly contributed to the group’s relatively high impact toughness and lower hardness. In contrast, the microstructure of group 9, as displayed in Figure 11i, revealed a substantial presence of dense needle-shaped martensite, exhibiting a uniform and dense distribution. This microstructural configuration was a key factor contributing to the specimen’s excellent macro-hardness performance. However, despite this superior hardness, the impact toughness of group 9 remained relatively low, possibly attributed to the excessively rapid cooling rate during deep cooling, leading to uneven bainite precipitation. It is evident that under the same holding conditions with a duration of 12 h, an increase in the cooling rate resulted in a more uniform and dense martensitic structure, thereby enhancing hardness performance.

The microstructure of group 3, as illustrated in Figure 11c, displayed a fine and uniform distribution of needle-shaped bainite and lath-shaped martensite intricately intermixed with each other. Unlike the dominant presence of a single structure observed in the previous groups, this particular group exhibited a blend of these two structures, contributing to a relatively balanced macroscopic performance in terms of impact toughness and Rockwell hardness. However, when considering the overall performance, the specific properties of each structure were appropriately manifested. The microstructure of group 5, as illustrated in Figure 11e, was characterized by a relatively higher proportion of lath-shaped martensite and coarse bainite. In comparison to group 3, it exhibited improved hardness but relatively lower impact toughness. On the other hand, the microstructure of group 7, depicted in Figure 11g, revealed the presence of two distinct structures: lath-shaped martensite and coarse bainite. This dual structure significantly influenced the observed mechanical properties, leading to enhanced hardness but diminished impact toughness in group 7. With increasing holding time, the material did not exhibit a preference for the precipitation of a specific type of bainite or martensite, resulting in a relatively balanced mechanical performance.

As previously mentioned, both shallow cryogenic treatment and deep cryogenic treatment exert an influence on the microstructure, subsequently impacting the macroscopic properties of the material. These different treatment processes give rise to variations in the morphology and distribution of martensite and bainite. In the present study, it was observed that with increased holding time, the martensitic structure becomes finer, and there is an interlocking arrangement of martensite and bainite, leading to exceptional overall performance. Previous research has demonstrated that low-temperature treatment enhances material performance by facilitating the transformation of austenite into martensite and promoting the precipitation of carbides [40]. In the study conducted by Zhang Haidong, it was highlighted that the deep cryogenic temperature has the most significant impact on the material’s wear resistance, with the maximum wear resistance achieved at −196 °C [20]. Similarly, D. Senthilkumar et al. pointed out that deep cryogenic treatment enhances the transformation of austenite to martensite more effectively than shallow cryogenic treatment [41]. However, in this present study, multiple performance characteristics of the material were taken into consideration, revealing that the holding time plays a more crucial role in overall performance enhancement. Interestingly, aligning with Zhang Haidong’s research, the optimal treatment temperature identified in this study was also −196 °C. After undergoing deep cryogenic treatment, a notable transformation occurs in the morphology of the martensitic structure. The findings of this study indicate that an increase in holding time leads to the transformation of the initial short and coarse martensite into finer and elongated plate-like structures. This observation aligns with the results reported in a previous study [42], where wide martensitic bundles were transformed into plate-like martensitic bundles after deep cryogenic treatment, resulting in improved hardness and strength of the material. The cryogenic treatment process also induces the precipitation of carbides within the martensitic structure. It is important to note that the reduction in carbon content within the martensite can potentially affect its hardness adversely. However, after cryogenic treatment, the martensitic structure becomes more stable, facilitating the effective dispersion of carbon atoms within the parent metal. This, in turn, enhances the bonding strength between carbon and iron atoms, leading to a positive effect on hardness [20,43]. Given the counteracting effects of these factors, the material’s hardness experiences minimal changes, which may explain the relatively unchanged hardness observed in this study. Furthermore, this study also investigated the impact of cryogenic treatment on the bainitic structure. It was observed that an increase in the bainitic content had a positive effect on the material’s impact toughness, enhancing its ability to withstand sudden loading or impact [44]. In the optimal treatment group identified in this study, a balanced distribution of both martensitic and bainitic structures was achieved, without a singular focus on the precipitation of a specific structure. This comprehensive approach contributed to an overall improvement in the material’s mechanical properties.

Figure 12 depicts the control group of the experiment, wherein the specimens were subjected to conventional heat treatment without any cryogenic treatment. The microstructure distribution in this group appeared relatively uniform, characterized by the presence of bainite and lath-shaped martensite arranged in regional patterns. The martensite exhibited a densely packed structure with a higher quantity, resembling the martensite distribution observed in the optimal test group. However, the microstructure in the control group exhibited shorter martensite formations, and the material did not exhibit superior hardness. This can be attributed to a reduced presence of bainite, as the material exhibited a preference for martensite precipitation, resulting in a decrease in the quantity of bainite due to enhanced martensite transformation [45]. Consequently, the impact toughness of the material in the control group was lower compared to that of group 3.

Transmission electron microscopy analysis was performed on specimens from group 3 and the control group. As shown in Figure 13a, the martensite structure in the control group was distributed in a lathed shape, with a relatively large and regular arrangement of carbide particles on the martensite surface. After cryogenic treatment, the martensite structure in group 3 became finer and shorter and underwent fragmentation with noticeable differences in orientation. The carbide particles became smaller and more evenly distributed, as shown in Figure 13b. During cryogenic treatment, the martensite lattice contracted, generating significant internal stress and increased crystal defects. Under the action of strain energy, supersaturated carbon atoms and alloying elements tended to aggregate at defects, forming new subgrain boundaries with carbon atoms, leading to the refinement of the martensite lath structure [21,46]. In subsequent tempering, the activity of carbon atoms increased, and the carbon atoms aggregated at defects in the form of carbide particles. As a result, the number of carbide particles increased, and the newly formed particles were smaller, thereby improving the wear resistance and hardness of the material.

### 3.5. Analysis of Impact Fracture

#### 3.5.1. Macroscopic Analysis of Impact Fracture

Ideally, the fracture surface of an impact specimen is composed of three parts: the fibrous zone, the radiative zone, and the shear lip zone. Under the same test conditions, the larger the proportion of the fibrous zone and the shear lip zone, the better the material’s impact toughness. As shown in Table 5 and Table 7, when considering only impact toughness, group 6 had the best performance relative to the control group. However, when considering both Rockwell hardness and wear resistance, group 3 had the best performance. Therefore, this section mainly analyzes the macroscopic fractures of group 6, group 3, and the control group. Figure 14 shows the macroscopic fracture surface of each group of specimens.

Figure 14a presents the fracture surface of the specimen from group 6, which exhibited a relatively flat morphology without a distinct radial region at the center. The majority of the fracture surface consisted of the fiber region, while crescent-shaped shear lips were observed on both sides. The presence of a significant proportion of shear lips and fiber regions indicates that the material exhibited exceptional impact toughness. In contrast, Figure 14b depicts the macroscopic fracture surface of the specimen from the control group. It displayed a distinct stepped shear surface with a visible radial pattern, albeit in a limited area. The fiber region constituted the largest portion, with shear lips only observed on one side. Turning to Figure 14c, representing the specimen from group 3, its fracture surface was not as flat as that of group 6. It exhibited some protrusions and a minor amount of radial patterning. The proportion of shear lips and fiber regions was higher compared to the control group, suggesting superior impact toughness relative to the control group but lower than that of group 6.

#### 3.5.2. Microscopic Analysis of Impact Fracture

According to the results presented in Table 5, the number of cryogenic treatments was identified as the most influential process parameter impacting the impact toughness of the 51CrV4 spring steel. Consequently, the microscopic fracture surface images of the specimens were categorized into three groups based on the number of cryogenic cycles: one time, two times, and three times. Subsequently, each group was subjected to individual analysis.

The morphology of the fracture surface in group 1 is depicted in Figure 15a. Clear dimples were observed on the fracture surface, accompanied by deeper secondary cracks. The dimples appeared to be small and shallow, indicating that the specimen in this treatment condition had a limited capacity for absorbing impact energy [47]. The generation of secondary cracks indicates that the specimen underwent significant deformation under the external impact, which also indicates that the specimen had poor toughness. Clear dimples were observed, suggesting a ductile fracture mechanism for the specimen under this process [48]. Figure 15b presented the fracture morphology of the specimen from group 5. Numerous dimples were observed on the fracture surface, exhibiting variations in their diameters and depths. Additionally, there were particle-like inclusions within these dimples, serving as stress concentration points and sources of crack propagation during impact fracture. This contributed to the lower impact toughness of the material [49]. The fracture mechanism of the specimen in this group was identified as a ductile fracture. Figure 15c showcased the fracture morphology of group 9. Medium-sized ductile dimples were observed on the fracture surface, also containing particle-like inclusions. However, the size of inclusions in this group was significantly larger than that of those in group 5. The larger size of inclusions resulted in greater stress concentration when subjected to external loads, leading to inferior impact toughness. The fracture mechanism of this group was identified as a ductile fracture.

Figure 16a illustrates the fracture morphology of group 2. The fracture surface displayed large equiaxed dimples, some of which were surrounded by inclusions. Most dimples had significant diameters and depths, indicating that the specimen absorbed a substantial amount of impact energy and exhibited good toughness under external impact loads [50]. The fracture mechanism in this group was classified as ductile fracture. In Figure 16b, the fracture morphology of group 6 was depicted. The dimples were evenly distributed with similar sizes, and a majority of them had large diameters and depths. At the bottom of some dimples, small particle-like inclusions were observed. The specimens in this group exhibited excellent impact toughness, with the fracture mechanism also identified as ductile fracture. Figure 16c presented the fracture surface of group 7, which displayed two distinct fracture characteristics. Sparse shallow dimples with small diameters were scattered in some corners, indicating relatively poor impact toughness. Moreover, noticeable stepped features were observed on the fracture surface, indicating cleavage characteristics. Therefore, the fracture mechanism of this group of specimens encompassed both ductile and brittle fractures.

Figure 17a depicts the fracture surface of group 3. The fracture surface displayed a dense distribution of small-diameter and shallow-depth toughness dimples, with occasional tiny inclusions present in some of the dimples. In comparison to the specimens from the other two groups, this specimen exhibited lower toughness. The fracture mechanism of this group of specimens was classified as ductile fracture. Figure 17b showcases the fracture surface of group 4. The fracture surface of this specimen revealed evenly distributed and dense equiaxed large-size toughness dimples, with slightly deeper depths. Some of the dimples contained small particle-shaped inclusions, indicating good impact toughness. The fracture mechanism of this group of specimens also belonged to ductile fracture. Figure 17c presents the fracture morphology of group 8. Within the microscopic range of the fracture surface, uniformly distributed large-sized equiaxed dimples with relatively deep depths could be observed. Only a few dimples were encapsulated with small-sized particle-shaped inclusions. The sample exhibited excellent impact toughness. The fracture mechanism of this group was identified as ductile fracture.

The fracture surface of the control group specimens is presented in Figure 18. When compared to the fracture surfaces of the specimens subjected to cryogenic treatment as described earlier, the number and size of dimples were reduced, the depths were shallower, and the shapes were more irregular. Additionally, multiple regularly arranged secondary cracks were observed on the fracture surface of this group of specimens. These findings suggest that the cracks propagated smoothly under external load, leading to fracture and lower impact toughness compared to the cryogenically treated specimens.

Overall, as the number of cycles increases, the fracture surface of 51CrV4 spring steel tends to exhibit larger equiaxed dimples with deeper depths, along with smaller inclusion sizes within the dimples. These observations indicate a reduction in defects and crack initiation points within the specimens. Consequently, the specimens become more durable and possess stronger impact resistance. This improvement can be attributed to the transformative effect of multiple cryogenic treatments on the material’s microstructure and the refinement of its internal organization, leading to enhanced impact toughness. These experimental findings are consistent with the work of Peng, J. [51]. It is noteworthy that when inclusions are larger in size, the fracture behavior demonstrates a ductile fracture. However, the presence of fan-shaped or river-shaped patterns of inclusions within the dimples indicates a brittle fracture. The occurrence of inclusions within the dimples results in stress concentration and has a detrimental impact on the material’s impact toughness.

EDS tests were performed on samples from group 3, group 6, and the control group, as illustrated in Figure 19. The analysis revealed that the particulate inclusions contained abundant elements such as Al, Ca, S, and Mg, suggesting their composition as SO_2_ and Al_2_O_3_-Mg-CaO composite oxides. The presence of these inclusions contributed to the formation of voids within the material. As depicted in Figure 19a, the impact fracture of the specimen from group 3 reveals the presence of inclusions rich in sulfur (S) element. It is well known that sulfur has a tendency to segregate at grain boundaries, resulting in a decrease in interfacial energy and promoting crack propagation along these boundaries. This phenomenon has a detrimental effect on the material’s impact toughness [52]. The energy spectrum analysis of the samples in group 6 revealed the presence of substantial amounts of aluminum (Al), calcium (Ca), sulfur (S), and magnesium (Mg) elements. Notably, the concentration of Al in the inclusions of this group was significantly higher compared to the concentration of S, while the content of S was similar to that of Ca within the inclusion spectra of these samples. Previous studies have demonstrated the positive influence of calcium (Ca) on impact toughness [53]. In this study, it was observed that higher levels of sulfur (S) content were associated with inferior impact toughness. However, as the relative concentration of Ca increased in comparison to S, the impact toughness improved. Therefore, it is speculated that the mitigating effect of Ca on the negative influence of S on impact toughness may be the underlying mechanism. For the selected test point in the control group, the EDS test results revealed the absence of sulfur (S) content, and the majority of the dimples were devoid of inclusions. These findings suggest that the deep cryogenic treatment may have facilitated the formation of dimples in 51CrV4 spring steel. Furthermore, with an increasing number of cryogenic cycles, the diameter and depth of the dimples on the fracture surface exhibited growth, while the S content within the inclusions within the dimples decreased. Consequently, these observations contribute to the enhanced impact toughness of the sample.

## 4. Conclusions

This study focuses on the investigation of 51CrV4 spring steel, employing the Taguchi method to analyze the impact of cooling rate, cryogenic temperature, holding time, and number of cycles on the material’s Rockwell hardness, impact toughness, and wear resistance. Through a comprehensive consideration of various material properties and utilizing gray correlation analysis, an optimal process group for the experiment was identified. Under this specific set of processes, the overall performance of 51CrV4 spring steel was significantly enhanced, providing valuable guidance for the manufacturing of leaf springs. The main conclusions derived from this study are presented as follows:(1)Considering the overall performance of 51CrV4 spring steel in terms of Rockwell hardness, impact toughness, and wear resistance, the holding time has the most significant influence on the comprehensive performance, with a contribution rate of 49.01%. The optimized process obtained through gray correlation analysis is as follows: a holding time of 24 h, a cooling rate of 1 °C/min, two cycles, and a cryogenic temperature of −196 °C.(2)Following cryogenic treatment, specimens from group 3 of 51CrV4 spring steel, characterized by a 24-h holding time, cooling rate of 1 °C/min, three cycles, and cryogenic temperature of −196 °C, exhibit notable improvements in various mechanical properties. Specifically, a substantial reduction of approximately 43.32% in wear is observed, while the Rockwell hardness shows a slight improvement compared to the control group, maintaining its level of hardness. Furthermore, a notable enhancement of 4.01% in impact toughness relative to the control group is achieved, along with an increase in overall strength.(3)Following cryogenic treatment, the microstructure of 51CrV4 spring steel exhibits distinct features, including well-defined elongated needle-shaped bainite and plate-like martensite. Additionally, small carbide particles precipitate, while coarse martensite fragments and noticeable variations in orientation occur. These microstructural alterations are macroscopically manifested through enhancements in impact toughness, hardness, and the wear resistance properties of the material.(4)With an increasing number of cryogenic treatments, the diameter and depth of equiaxed tough areas on the fracture surface of 51CrV4 spring steel exhibit a notable growth. Simultaneously, the size of inclusions within these tough areas diminishes, indicating reduced specimen defects and enhanced impact toughness. Furthermore, the impact of elements present in the inclusions on toughness was explored through EDS analysis. The findings reveal a positive influence of Ca on impact toughness, while S exhibits a negative effect.

## Figures and Tables

**Figure 1 materials-16-04399-f001:**
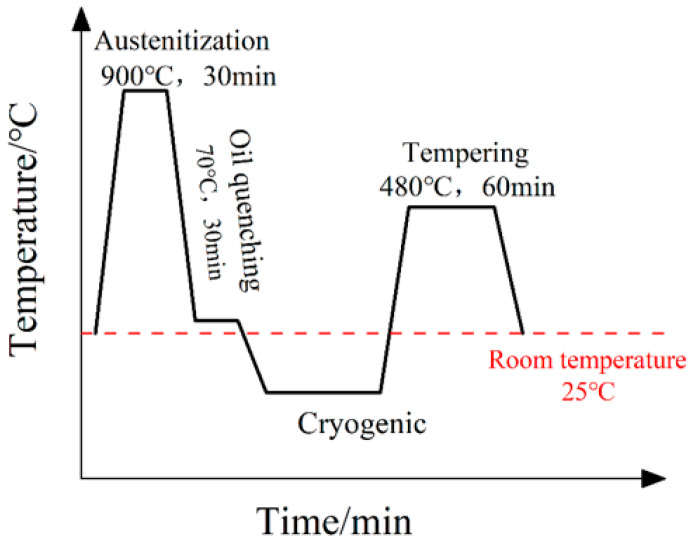
Heat treatment process used in the experiment.

**Figure 2 materials-16-04399-f002:**
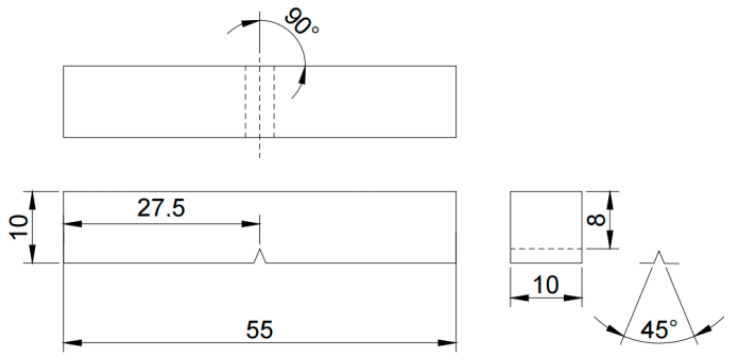
Dimensional drawing of the impact specimen (mm).

**Figure 3 materials-16-04399-f003:**
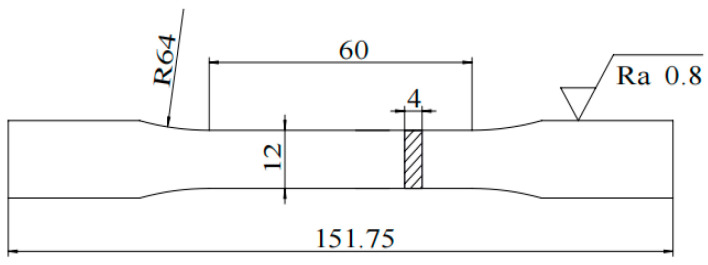
Dimensional drawing of tensile specimen.

**Figure 4 materials-16-04399-f004:**
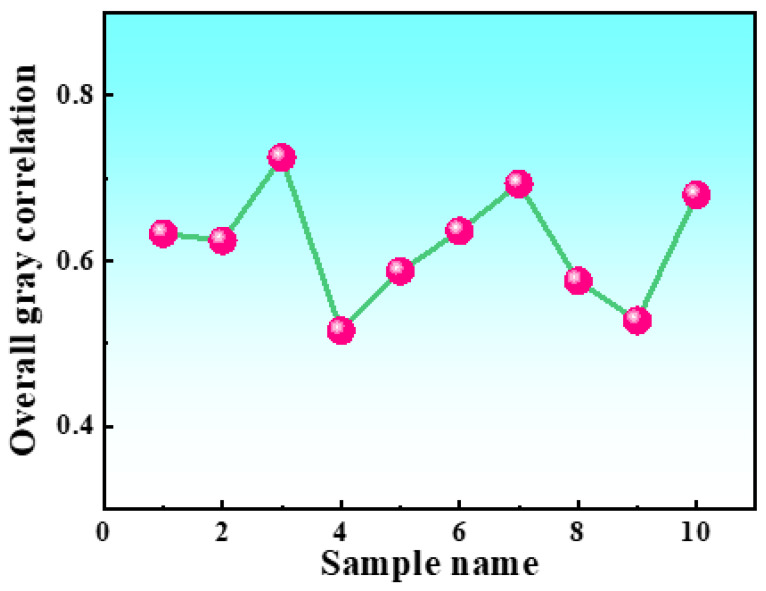
Overall gray correlation chart.

**Figure 5 materials-16-04399-f005:**
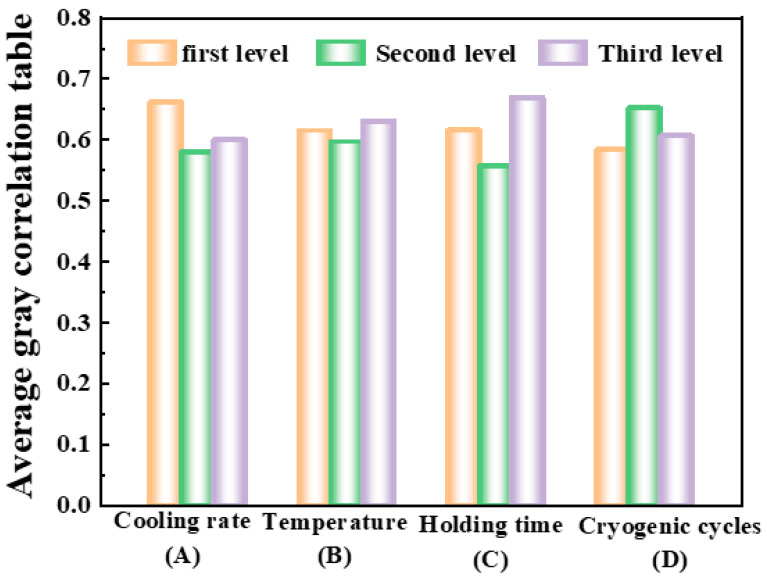
Average gray correlation chart.

**Figure 6 materials-16-04399-f006:**
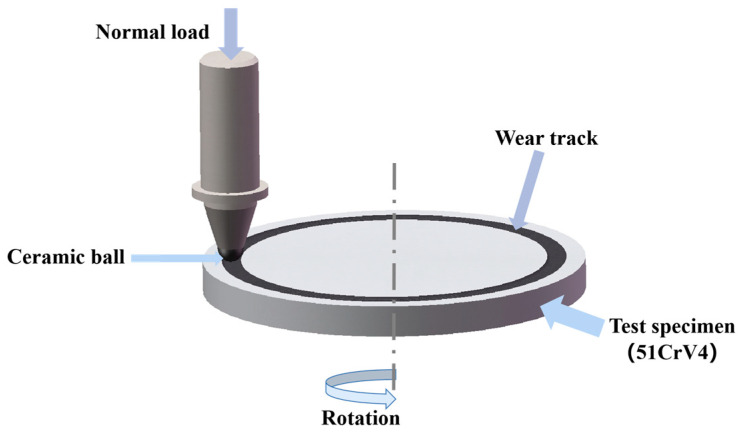
Schematic diagram of the friction system.

**Figure 7 materials-16-04399-f007:**
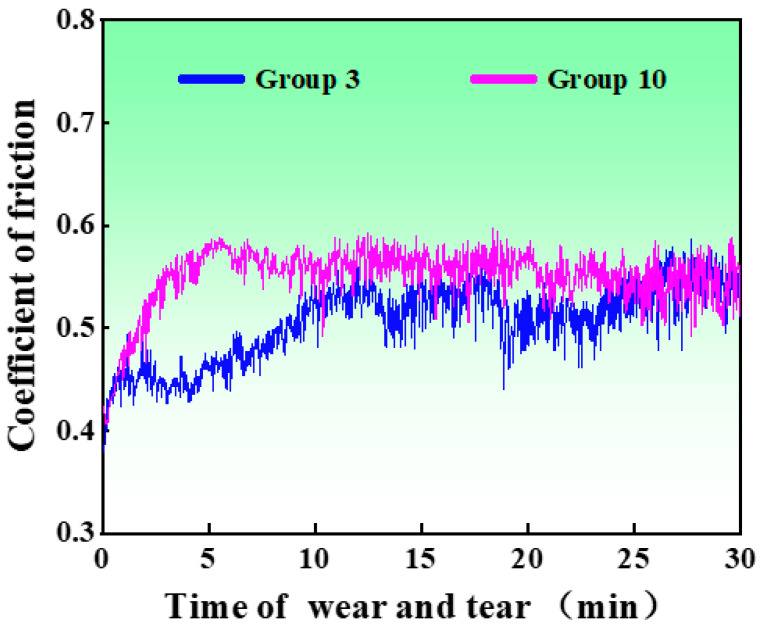
Instantaneous friction coefficient.

**Figure 8 materials-16-04399-f008:**
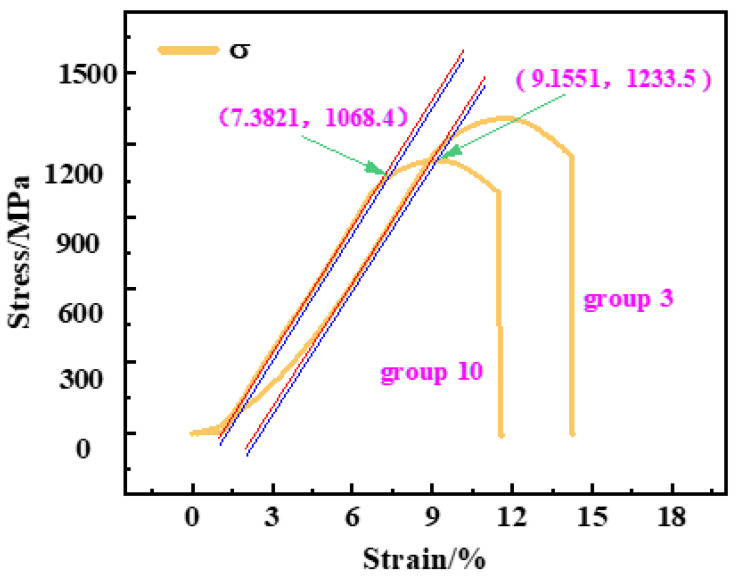
Stress–strain diagram after cryogenic treatment and conventional heat treatment.

**Figure 9 materials-16-04399-f009:**
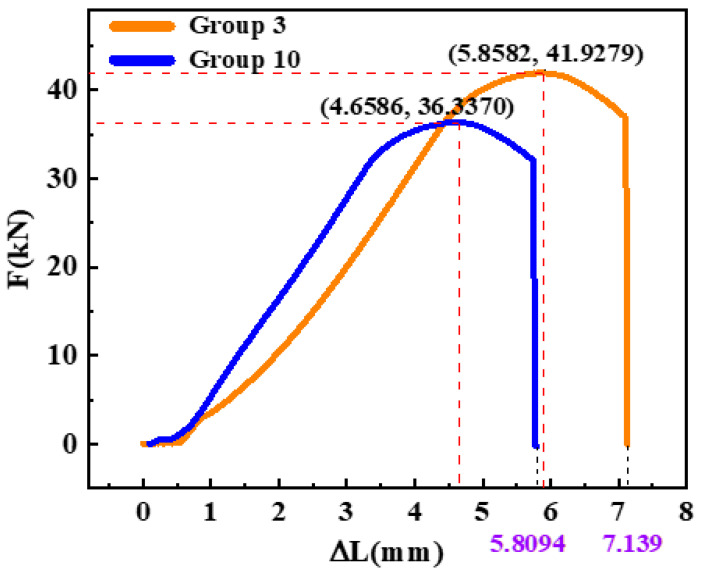
Force: the length of the increment diagram.

**Figure 10 materials-16-04399-f010:**
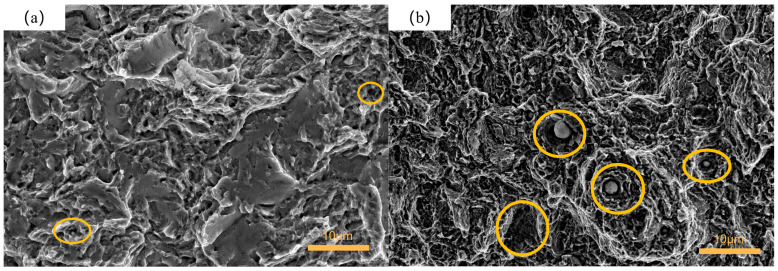
SEM of tensile fracture: (**a**) control group; (**b**) group 3.

**Figure 11 materials-16-04399-f011:**
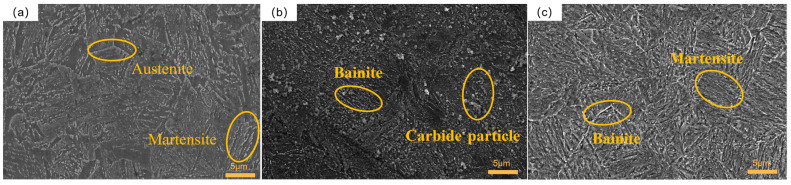
SEM of specimens for different cryogenic processes: (**a**) group 1; (**b**) group 2; (**c**) group 3; (**d**) group 4; (**e**) group 5; (**f**) group 6; (**g**) group 7; (**h**) group 8; (**i**) group 9.

**Figure 12 materials-16-04399-f012:**
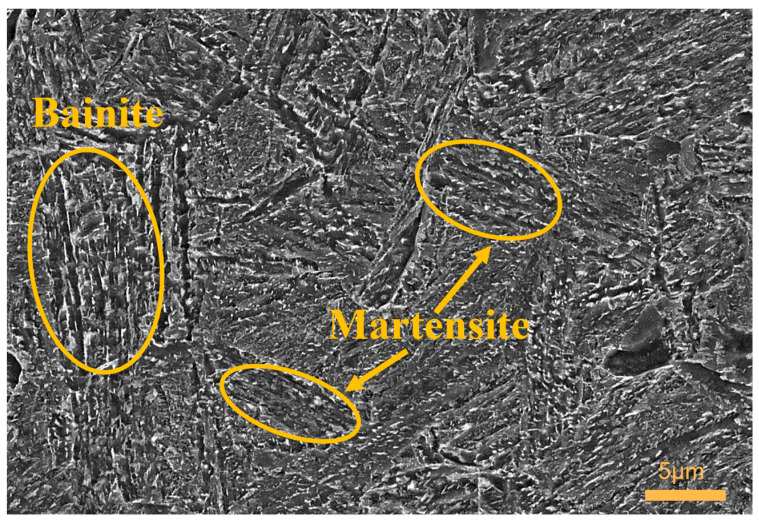
Microstructure of control group specimens.

**Figure 13 materials-16-04399-f013:**
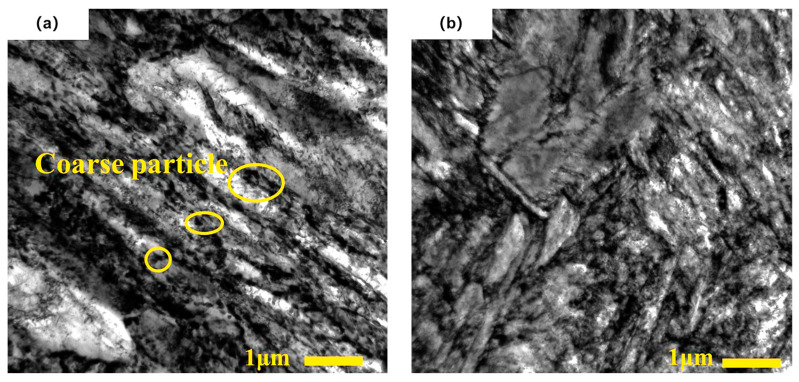
TEM diagram of specimens for different cryogenic processes: (**a**) control group; (**b**) group 3.

**Figure 14 materials-16-04399-f014:**
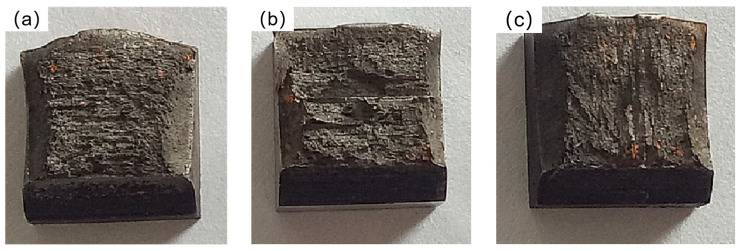
Macroscopic fracture morphology of the specimen: (**a**) group 6; (**b**) control group; (**c**) group 3.

**Figure 15 materials-16-04399-f015:**
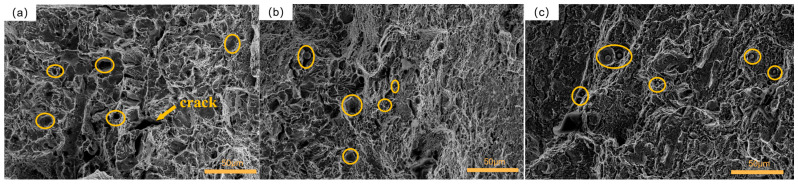
Fracture morphology of the samples with 1 cryogenic cycle: (**a**) group 1; (**b**) group 5; (**c**) group 9.

**Figure 16 materials-16-04399-f016:**
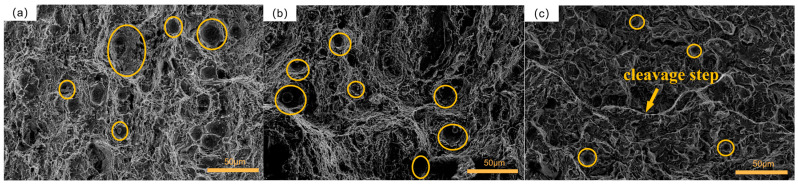
Fracture morphology of the samples with 2 cryogenic cycles: (**a**) group 2; (**b**) group 6; (**c**) group 7.

**Figure 17 materials-16-04399-f017:**
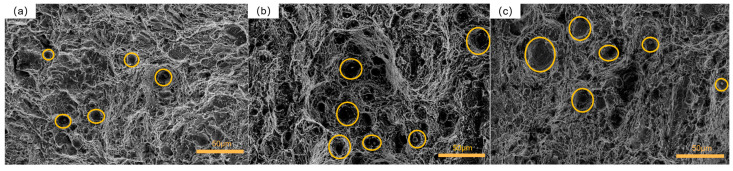
Fracture morphology of the samples with 3 cryogenic cycles: (**a**) group 3; (**b**) group 4; (**c**) group 8.

**Figure 18 materials-16-04399-f018:**
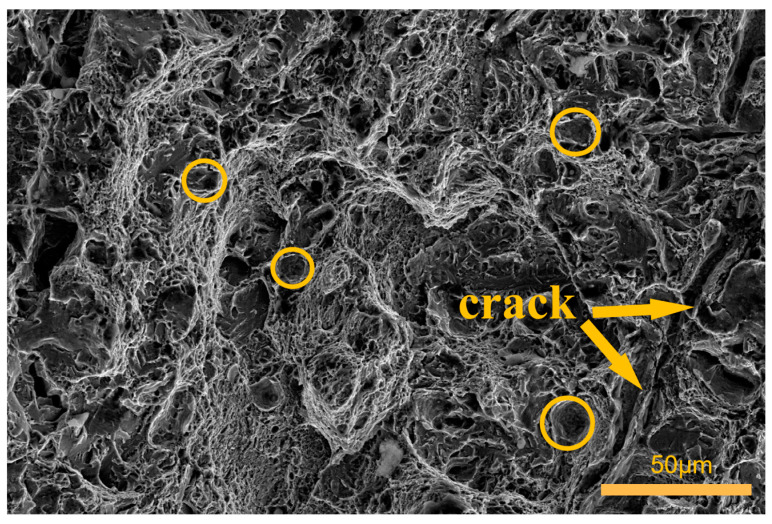
Fracture morphology of specimens in the control group.

**Figure 19 materials-16-04399-f019:**
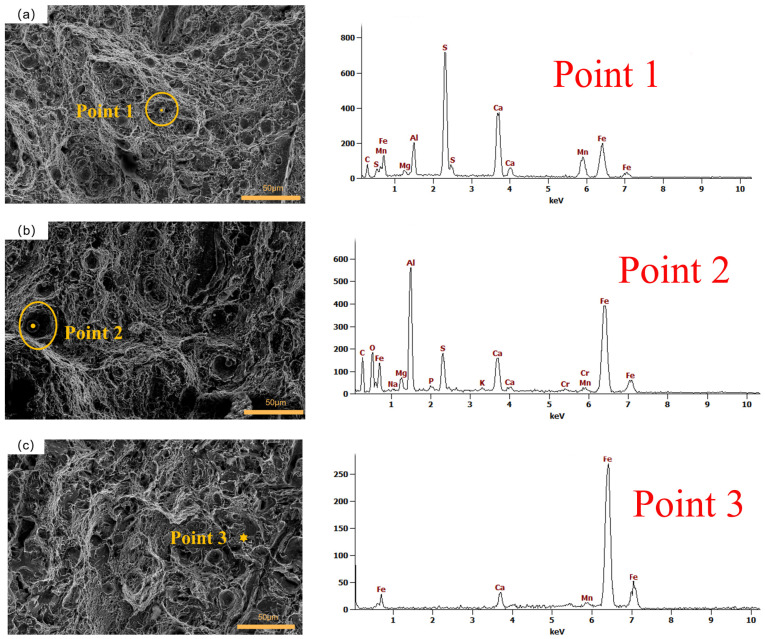
EDS tests for impact fracture: (**a**) group 3; (**b**) group 6; (**c**) control group.

**Table 1 materials-16-04399-t001:** Chemical composition of 51CrV4 spring steel w(t)%.

Element	Mass Percentage	GB/T 1222-2007
C	0.48	0.46–0.54
Si	0.3	0.17–0.37
Mn	0.71	0.5–0.8
S	0.025	≤0.025
P	0.025	≤0.025
Cr	1.02	0.8–1.1
Ni	0.03	≤0.035
Cu	0.02	≤0.025
V	0.12	0.1–0.2
Fe	balance	

**Table 2 materials-16-04399-t002:** Cryogenic treatment test program.

Sample Name	Cryogenic Treatment			
	Cooling Rate/K·min^−1^	Temperature/°C	Holding Time/h	Cryogenic Cycles
1	1	−120	2	1
2	1	−160	12	2
3	1	−196	24	3
4	3	−120	12	3
5	3	−160	24	1
6	3	−196	2	2
7	5	−120	24	2
8	5	−160	2	3
9	5	−196	12	1
10	-	-	-	-

**Table 3 materials-16-04399-t003:** Test parameters of CFT-1 type normal temperature tribometer.

Load/N	Speed/r·min^−1^	Radius/mm	Time/min
90	1500	6	30

**Table 4 materials-16-04399-t004:** Test results.

Sample Name	Factors				Performance Metrics		
	Cooling Rate (K/min)	Temperature (°C)	Holding Time (h)	Cryogenic Cycles	Wear Mass Loss (mg)	Hardness (HRC)	Impact Toughness (J/cm^2^)
1	1	−120	2	1	4.788	38.52	36.91
2	1	−160	12	2	7.933	31.97	77.01
3	1	−196	24	3	6.866	42.55	48.71
4	3	−120	12	3	51.112	33.22	87.00
5	3	−160	24	1	13.308	39.38	42.89
6	3	−196	2	2	29.395	30.11	97.95
7	5	−120	24	2	17.202	43.75	41.63
8	5	−160	2	3	49.522	31.33	96.23
9	5	−196	12	1	71.397	43.02	40.61
10	-	-	-	-	12.113	42.51	46.83

**Table 5 materials-16-04399-t005:** Range analysis table.

Level	Wear Mass Loss (mg)				Hardness (HRC)				Impact Toughness (J/cm^2^)			
	A	B	C	D	A	B	C	D	A	B	C	D
1	19.59	73.10	83.71	89.49	113.04	115.49	99.96	120.92	162.63	165.54	231.09	120.41
2	93.82	70.76	130.44	54.53	102.71	102.68	108.21	105.83	227.84	216.13	204.62	216.59
3	138.12	107.66	37.38	107.50	118.10	115.68	125.68	107.10	178.47	187.27	133.23	231.94
Rx	118.53	36.90	93.07	52.97	15.39	13.00	25.72	15.09	65.21	50.59	97.86	111.53
Rank	1	4	2	3	2	4	1	3	3	4	2	1

**Table 6 materials-16-04399-t006:** Normalized processing data table.

Sample Name	Performance Metrics		
	Wear Mass Loss	Hardness	Impact Toughness
1	1.000000	0.61657	0.00000
2	0.952784	0.13636	0.65695
3	0.968803	0.91202	0.19332
4	0.304538	0.22801	0.82061
5	0.872089	0.67962	0.09797
6	0.630575	0.00000	1.00000
7	0.813629	1.00000	0.07733
8	0.328409	0.08944	0.97182
9	0.000000	0.94648	0.06062
10	0.890030	0.90909	0.16252

**Table 7 materials-16-04399-t007:** Gray correlation coefficient and overall gray correlation scale for each performance.

Sample Name	Factors				Gray Relational Coefficient ( γ = 0.5)			Overall Gray Correlation	Rank
	A	B	C	D	Wear Mass Loss	Hardness	Impact Toughness		
1	1	−120		2	1.00000	0.56598	0.33333	0.63310	5
2	1	−160		12	0.91372	0.36667	0.59308	0.62449	6
3	1	−196		24	0.94127	0.85037	0.38265	0.72476	1
4	3	−120		12	0.41825	0.39308	0.73595	0.51576	10
5	3	−160		24	0.79629	0.60947	0.35663	0.58746	7
6	3	−196		2	0.57509	0.33333	1.00000	0.63614	4
7	5	−120		24	0.72847	1.00000	0.35145	0.69331	2
8	5	−160		2	0.42677	0.35447	0.94665	0.57596	8
9	5	−196		12	0.33333	0.90331	0.34737	0.52800	9
10	-	-		-	0.81971	0.84615	0.37384	0.67990	3

**Table 8 materials-16-04399-t008:** Average gray correlation.

Level	Factors			
	Cooling Rate (A)	Temperature (B)	Holding Time (C)	Cryogenic Cycles (D)
1	0.66078	0.61406	0.61507	0.58286
2	0.57979	0.59597	0.55609	0.65131
3	0.59909	0.62964	0.66851	0.60550
4	0.08100	0.03367	0.11243	0.06846
5	2	4	1	3

**Table 9 materials-16-04399-t009:** Overall gray correlation analysis of variance.

Factors	Degrees of Freedom	Sum of Squares	Mean Square Sum	F-Value	Contribution Ratio
Factor A	2	0.01074	0.00537	6.30609	27.74%
Factor C	2	0.01898	0.00949	11.14222	49.01%
Factor D	2	0.00730	0.00365	4.28591	18.85%
Error	2	0.00170	0.00085		
Total	8	0.03872			

## Data Availability

All data that support the findings of this study are included within the article.

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
