# Peer review of "Impact of Cryogenic Treatment Process on the Performance of 51CrV4 Steel"

_materials, 2023, doi:10.3390/ma16124399_

Round 1

Reviewer 1 Report

The results presented in this article provide a comprehensive investigation of the effects of cryogenic treatment on the properties of 51CrV4 spring steel. The use of the Taguchi method and gray correlation analysis allowed for the selection of an optimized process group that could improve the overall performance of the material. The results indicate that the holding time has the most significant influence on the comprehensive performance of the material.

Authors can consider the following comments.

Decision: minor revision

(1) In abstract section, In the following sentence, it is understood as different researchers apart from the authors.

“ Researchers used cryogenic ….”

(2) For section 3.2.:

I would suggest that the authors could provide more detail on the limitations of the study. While the paragraph acknowledges the limitations of the study in terms of its analysis of friction coefficients, it would be helpful if the authors could elaborate on these limitations and discuss how they might impact the interpretation of the study's results.

Additionally, I would encourage the authors to consider the broader implications of their findings and to discuss how they might be relevant to other industries or applications beyond the specific one considered in this study.

(3) For section 3.4:

I want to ask the authors to explain why the presence of residual austenite in the first cryogenic treatment group resulted in lower hardness and impact toughness.

(4) For Figure 9, arrows can be marked on the SEM image to highlight different structures.

Reviewer 2 Report

The present study investigated the impact of the cryogenic treatment process on the performance of 51CrV4 Steel. The paper is obviously of interest to researchers working in this field. However, the manuscript should be amended before its acceptance. I want to address the following issues:

1. Table 1 shows the chemical composition of 51CrV4 steel. According to the reference data, the steel has an excessively high Ni and Cu content (0.3% and 0.2%, respectively). Please clarify.

2. In Experimental Procedure, more details of the tribological tests should be included. There needs to be a description (figure) of the tribological system. There needs to be more information on what material was counter-tested, etc.

3. The term "Wear quality" is used in Table 4. This is an incorrect term. Only the weight loss during tribological tests was determined. The comment applies to the entire article.

4. In Fig. 7, please remove the word "engineering" from the x and y axes and present the results in MPa. It needs to be described what the red and blue lines represent. The reader can only guess.

5. In Fig. 8, the unit of force should be corrected. It is "KN" and should be "kN".

6. The units in the figures are represented differently, e.g. in Figure 7 "/GPa" and Figure 8 (KN). Please keep the units written uniformly throughout the article.

7. The quality of some of the microscope figures could be better. When zoomed in, details are not visible.

8. The results obtained in the paper are not correlated with other scientific publications. Please also refer to the work of other scientists in this area.

Reviewer 3 Report

1.      Abstract should be given as more interesting. Express at least one of the main aspects and features of the paper.

2.      The abstract should highlight the findings of this work, such as the numeric values.

3.      Improve the conclusion part of the Abstract.

4.      Research gap studies are need to be elaborate.

5.      Wherever applicable, the scientific explanation needs to be added and the research novelties need to be clearly emphasized.

6.      At the end of Introduction section, it would be better to add the paper's organization in different paragraph

7.      There is no scientific justification for the selection of factors and range of factors for cryogenic treatment.

8.      Given that the manuscript is based on experimental work and measurements, it is vital for the authors to report on the method(s) to improve measurement reliability. The methods/measures that have been taken to minimize experimental errors and improve reliability should be included in the experimental setup.

9.      Further, results and analysis of experiments should be compared with previous researchers by citing references.

10.  Manuscript must be presented in highlight the contribution, and applicability of the work.

11.  Please check the manuscript for wrong choice of words, grammatical errors and incoherent sentence structure.

12.  Future scope should be clearly written in the conclusion section.

Please check the manuscript for wrong choice of words, grammatical errors and incoherent sentence structure.

Reviewer 4 Report

Dear Authors!

If you want to use the taguchi method to fined the best parameters for the cryogenis or deep cryogenic treatment you should chose another steel.  Cryogenic treatments are used for high alloyed tool steels to minimize the residul austenite phases after quenching. In these steels the amount of residual austenite after quenching can be as high as 22-26%. In the case of 51CrV4 spring steel the formation of resiual austenite is neglijable, so it makes no sense to use the cryogenic treatments. In your manuscript you use different cryogenic heat teatments temperatures (-120; -160; -196 °C), in the technical literature between 0 °C and -80°C the correct technical word is sub zero treatment, between -81 and -149°C are used shallow cryogenic treatment and between -150--196°C is deep cryogenic treatment.  I am interested what type of heat treatment furnace do you used , how you controlled the temperature, the heating and cooling rate. 

I do not understand while was investigated the wear resistance and the impact toughness for this steel, spring steels are not developed for such properties. 

Another thing is that multiple cycles are not used in cryogenic treatment practice. 

The images from figures 10,11, 12, 14,15,16,17 and 18 are the por quality, are unintelligible and unreadable.

Please describe clearly the novelity that your manuscript represents for the scientific community and what is the practical motivation behind the developement of your work. Are there any industrial application on this work?

The technical English language need to be improwed.

Reviewer 5 Report

Unfortunately, the manuscript does not meet the required standards for an article to be published in a given journal. The quality and novelty of the manuscript is low. The literature review is not sufficient for the spring steels, all the other steels are mentioned, but not the selected group of steels. The use of the DCT method is also highly questionable. In addition, the discussion section does not include references to previous literature as such, as this is not the first study to observe the described phonemona. For these reasons, it is proposed that the manuscript be rejected.

Extensive English editing required due to formatting and writing style.

Round 2

Reviewer 3 Report

Authors have made significant changes in the revised manuscript. Hence, consider the manuscript for publication in its present form.

Author Response

Dear Reviewer:

Thank you very much for taking your valuable time to review and for your approval of the manuscript.

Reviewer 4 Report

Accept the revised manuscript.

I accept the revised Quality of English Language.

Author Response

(The authors gave the same response as above.)

Reviewer 5 Report

Despite the opportunity given to the authors to revise the manuscript, and then the authors revising the manuscript, the manuscript still lacks scientific soundness and understanding of the fundamental difference between DCT and SCT (Shallow Cryogenic Treatment). The temperature of -120C is under SCT and not DCT. This clearly shows the lack of knowledge on the subject. If the literature review was better and more detailed and not biased, the design of the Taguchi method and then the experiments would be carried out within DCT temperatures. In addition, the authors do not only test DCT, but also parameters where some research has been done, but no references were given on this topic (Darwin et al, Das et al, Pellizzari et al, Jovicevic-Klug et al, etc.). Furthermore, the discussion section still does not include references to previous literature as such, since this is not the first study to observe the described phonemona. The phenomena have been observed in other similar steels with similar chemical composition or microstructure. On this basis, the decision to reject the manuscript remains.

The extensive English corrections must be made to make the manuscript readable and to ensure that all the notations are correct.
